# Measurement of Water Vapor Condensation on Apple Surfaces during Controlled Atmosphere Storage

**DOI:** 10.3390/s23031739

**Published:** 2023-02-03

**Authors:** Manfred Linke, Ulrike Praeger, Daniel A. Neuwald, Martin Geyer

**Affiliations:** 1Department of Horticultural Engineering, Leibniz Institute for Agricultural Engineering and Bioeconomy (ATB), 14469 Potsdam, Germany; 2Lake of Constance Research Centre for Fruit Cultivation (KOB), 88213 Ravensburg, Germany

**Keywords:** apple storage, wetness sensor, condensation retention time, dew point undershoot, atmospheric evaporation, postharvest

## Abstract

Apples are stored at temperatures close to 0 °C and high relative humidity (up to 95%) under controlled atmosphere conditions. Under these conditions, the cyclic operation of the refrigeration machine and the associated temperature fluctuations can lead to localized undershoots of the dew point on fruit surfaces. The primary question for the present study was to prove that such condensation processes can be measured under practical conditions during apple storage. Using the example of a measuring point in the upper apple layer of a large bin in the supply air area, this evidence was provided. Using two independent measuring methods, a wetness sensor attached to the apple surface and determination of climatic conditions near the fruit, the phases of condensation, namely active condensation and evaporation, were measured over three weeks as a function of the operating time of the cooling system components (refrigeration machine, fans, defrosting regime). The system for measurement and continuous data acquisition in the case of an airtight CA-storage room is presented and the influence of the operation of the cooling system components in relation to condensation phenomena was evaluated. Depending on the set point specifications for ventilation and defrost control, condensed water was present on the apple surface between 33.4% and 100% of the duration of the varying cooling/re-warming cycles.

## 1. Introduction

Apples can be stored for several months in controlled atmosphere (CA) conditions at temperatures between 0 and 4 °C. To prevent moisture loss, the relative humidity should be kept between 85% and 95% [1,2,3]. Some studies recommend even higher humidity levels of 92–95% [4].

An important criterion for long-term storage of apples is mass loss, which should not exceed 3.5% at the end of storage [5]. This mass loss is unavoidable since it is mainly caused by transpiration and respiration of the apples. While CA storage can limit the respiratory activity, the predominant transpiration losses are primarily dependent on humidity and airflow.

Apples are usually stored in large plastic bins (approx. 300 kg, dimensions 1.20 × 1.00 × 0.78 m) arranged close to each other, with up to 12 levels. This leads to different heat and mass transfer conditions within the storage room, mainly due to varying airflow [6,7,8]. This applies both to bins in different locations within the storage room and to the apples within single bins. Apples in the upper layer of a bin are exposed to more intense airflow compared to those located in the center and bottom of the bin. Even vents on the side walls of bins cannot fully compensate for these differences.

Cold storage often results in temperature fluctuations in the ambient air due to the operating cycle of the refrigeration system [9,10]. These fluctuations are most severe in locations where the cold air directly hits the fruit. The cyclical operating mode of the refrigeration system and the resulting temperature fluctuations, as well as the very high air humidity in the storage room, can lead to temperature drops below the dew point [11,12]. As a result, part of the water vapor present in the humid air is condensed in the form of individual droplets or film or a mixture of both on surfaces of solids, including apples. The amount and form of condensation mainly depends on the physical parameters and wettability of the surface, which is determined by the free surface energy of the wall material and the surface tension of the condensing fluid [13].

Condensed water on the surface of fruit and vegetables has negative effects on the external appearance, such as skin color, surface structure, and texture [14,15,16], and provides excellent conditions for the growth of microorganisms. Both condensate on the fruit surface and very high humidity in the environment, in combination with other parameters (e.g., temperature, pH, microbial preload), promote microbial activity and the associated negative effects on product quality [17,18,19]. Liquid water on the produce surface can potentially cause more deterioration than high relative humidity near the produce [20]. On the other hand, condensation on the fruit surface can also have positive effects on shelf life. As long as condensed water is present on the fruit surface, unwanted water loss due to transpiration is prevented [21].

Optimizing climate control during storage requires understanding the relationship between the cooling system operation, ventilation, defrosting time, temperature fluctuations, and fruit behavior. In this context, monitoring condensation on fruit surfaces is crucial because of its impact on heat and mass transfer.

Commercially available sensors are not suitable for measuring condensation on bulky fruit surfaces in cold storage. Capacitive and resistive sensors require a flat contact surface and are affected by the thermal conductivity of the carrier material [22]. Capacitive sensors that are based on the difference in the dielectric constant of water and air cannot measure small amounts of condensate on compact fruit with high water content [23,24]. Newer versions with hydrophilic coatings [25] are more suitable, but they require longer recovery times so rapid changes between wet and dry surfaces cannot be detected in apple storage.

Apples build up natural protective mechanisms during their development, including limiting excessive water losses [26,27,28]. For this purpose, waxy components are stored in the outer tissue layers [29,30]. The structure and composition of these protective layers depend on various internal as well as external conditions, such as variety, pre- and postharvest conditions, age, and maturity. Particularly due to wax components, condensation water droplets form on the surface of the fruit when its temperature falls below the dew point of the surrounding air. Such droplets develop at locations with small irregularities, e.g., lenticels and surface defects such as microcracks, and increase in size as more water vapor condenses on the exposed surface. Uneven wetting impairs resistive measurement of condensate on plant surfaces [31]. The surface tension of the water can be reduced by adding surface active agents (surfactants), thus achieving higher wetting of the surface [32].

A previous study [21] investigated the condensation processes on individual fruit with unrestricted free convection under laboratory conditions. Direct and indirect measurement methods were analyzed and discussed in order to record how long condensed water remained on the fruit surface and to distinguish the two phases of condensation, namely active condensation and atmospheric evaporation.

The objective of this study was to use a new wetness sensor [21] to detect condensation on apple surfaces during controlled atmosphere storage in a 50 t room. This is the first time condensation on the apple surface was measured over a period of 3–4 weeks under realistic heat and mass transfer conditions. To verify the direct measurement, the condensation phases were indirectly determined at the same time by measurement of the dew point undershoot. Additionally, the study aimed to evaluate the effects of the defrosting and ventilation regimes of the refrigeration system on condensation.

## 2. Materials and Methods

### 2.1. Storage Conditions

The ‘Jonagold’ apple cultivar was harvested from the Lake of Constance Research Centre for Fruit Cultivation (KOB) in Ravensburg, Germany and stored in a 50 t room for seven months under CA conditions (1.0 kPa O_2_ and 2.5 kPa CO_2_). The technology required to control the storage conditions (temperature and gas concentrations) was provided by Isolcell (Leifers, Italy). The storage room was filled to capacity with 163 standard plastic bins, each containing 300 kg fruit. The bins were stacked in three rows of bin stacks eight bins high. On the room’s back wall, below the ceiling, an air cooler with an oppressive air duct (Helpman LFX 256-7; Alfa Laval Corporate AB, Lund, Sweden) was attached, which contained evaporator and ventilation units (Figure 1). The air duct was fitted with five axial fans (ø 356 mm) for air circulation and had a maximum volumetric flow rate of 14,000 m^3^ h^−1^.

The control of the refrigeration system (DuDH5-150t; DWM Copeland Europe, Aachen, Germany) was load-dependent with a set point of 1.0 ± 0.5 °C. As a result, a constant alternation of cooling and re-warming phases occurred in the storage room. The fans were operated to optimize energy efficiency. Additionally, all process sections were superimposed by controlling the defrosting process on the evaporator, which worked according to a daily fixed time cycle.

When investigating the influence of the operating mode of the cooling system on the condensation phenomena, uniform phases of a cycle were defined:The cooling phase began with a delayed connection (2 min) of the refrigeration system after reaching or exceeding the upper air temperature setpoint (1.5 °C).The refrigeration system switched off when the lower air temperature setpoint (0.5 °C) was reached.The cycle ended after a re-warming phase when the upper temperature setpoint was reached again.

During the cooling phase, the fans operated continuously and for another 10 min after the desired temperature was reached (refrigeration system off). During the subsequent re-warming phase, the fans were turned off and on in a 40/20 min cycle per hour, which was either interrupted or ended by the next cooling phase or a defrosting process.

Control of the refrigeration system and fans was overlaid by a time control for defrosting the evaporator. For this purpose, five 40 min defrosting cycles (cooling off, fan 30 min on +10 min off) were implemented at fixed time points per day. The existing additional heating unit to support the defrosting of the evaporator was not used during the condensation measurements.

### 2.2. Experimental Design

Measurements were carried out in the supply air area in two bins at two levels, for a total of four bins. Two bins located in the middle and right-hand rows of the bin stacks in tiers 3 and 7 were used. Each bin contained a measuring point in the top layer of apples in direct contact with the cold supply air. Bins in the middle rows of tiers 3 and 7 had an additional measuring point located towards the center of the apple bin, 0.2 m below the top layer (Figure 1). A wetness sensor, an infrared surface temperature sensor, and an air temperature/humidity sensor were positioned at each of the six measuring points. The sensors were installed approx. 2 months after the start of storage with a short interruption of the CA storage. The measurement methods are described in Section 2.4.

To investigate the effects of the operating mode of the refrigeration system on the condensation processes on the fruit surface, the measuring point in the middle row (upper layer of apples in the bin) of tier 3 was selected (Figure 1). At this measuring point, both sensors for condensation measurement (infrared temperature and wetness) were attached to the same apple with an offset by approx. 90 degrees, which thus practically functioned as the measuring fruit (Figure 2). The “Granny Smith” apple variety was chosen for sensor application due to its long shelf life.

To better characterize the selected measuring point, air speed was measured in both the top layers between the apples in the bin and the free airflow approx. 0.10 m above the fruit layer. An in-house developed omnidirectional air speed logger (ASL) (ATB, Potsdam, Germany) was used to measure air movement in close vicinity to the apples [33]. In the free airflow at some distance from the fruit stored in the top layer, the air velocity was measured using a FVAD TH4 thermoanemometer (Ahlborn, Holzkirchen, Germany).

### 2.3. Data Acquisition System

A wired data acquisition system was installed on-site to accommodate local conditions. This system enabled recording the anticipated measurements in the storage room, which could not be taken during the measurements, and at the same time, visualizing the resulting measurements outside of the storage room. This system also enabled checking the functionality of the measured data acquisition and saving the measurements in real time. Two data acquisition systems were used for recording data from the six measuring points.

A data acquisition unit (Figure 3) consisted of two microprocessor systems (Arduino Mega 2560, Turin, Italy) communicating with each other as master or slave via an inter integrated circuit (I2C) bus. The I2C bus (e.g., length 6 m) was equipped with two bus amplifiers (Active I2C Long Cable Extender P82B715 Module; Philips, Amsterdam, The Netherlands) between master and slave. The master was positioned near the designated measurement location in the storage room and the slave outside the storage room. This positioning allowed the functionality of the system to be checked and so that the data could be saved on a Secure Digital (SD) card and read out every few days.

In the storage room, all sensors were connected to the master via 1.5 m long cables, which recorded all measured values at fixed time intervals (1 min) and sent the data to the slave via the I2C bus. Additionally, the slave was equipped with a data logger shield (SD card slot), a real-time clock (DS3231), and a 1.3 inch mini display, which received the measurement data and saved them together with the date and time on the SD card. Selected measured values and a dataset counter were shown on the display for control purposes. The dataset counter (DS) signaled that the data had been stored on the SD card.

Master and slave were connected via a 4-core, shielded cable. Two wires were used for the power supply and two wires were used for the I2C bus (serial clock, serial data). The slave supplied itself and all connected components, including the master, with power via its standard USB interface (5V). The entire data acquisition unit was put into operation by switching on the power supply and was switched off by disconnecting it from the power supply. Both microprocessor units were contained in hermetically sealed housings to keep moisture away from the sensitive electronic components.

The software created in C ++ was designed to acquire data from nine sensors (6 digital + 3 analog), transfer the data from the master to the slave, and store and display the data. Each of the digital sensors recorded two parameters (temperature + humidity or temperature + surface temperature). The measured data was transferred from the master to the slave in two data packets of 32 bytes each at a time interval of 1 min.

### 2.4. Condensation Measurement

Based on the selected sensor configuration, two independent measurement methods, wetness sensor and dew point undershoot, were implemented to detect particular aspects of condensation on the fruit surfaces under the given conditions [21]. The active phase of condensation could be determined by both methods. Thus, a possibility of checking the plausibility of the measurements was provided. Nevertheless, smaller measurement differences could not be excluded due to local flow conditions (convective mass transfer coefficients) since the measurements were carried out at different measurement locations on the apple surfaces [21].

Based on the combination of air temperature/humidity sensor and infrared surface temperature sensor, the active phase of condensation could be determined when the apple surface temperature fell below the dew point temperature of the surrounding air. In contrast, the wetness sensor signaled water on the surface due to both phases of the condensation process, active condensation and atmospheric evaporation. It was necessary to take into account the possibility that for a short period of time there may have been locally differentiated active condensation as well as atmospheric evaporation on the fruit [21]. Statements on the intensity of the condensation could only be made qualitatively using these two methods.

#### 2.4.1. Dew Point Undershoot

The surface temperature of the fruit, as well as the air temperature and relative humidity, were measured to determine if the temperature dropped below the dew point.

To measure the surface temperature of an apple, a contactless infrared thermometer (MLX 90416; Melexis Technologies, Tessenderlo, Belgium) was used. It consisted of an infrared sensitive thermopile detector chip and a signal conditioning chip integrated into a single unit. The low noise 17-bit ADC and signal-processing electronics were contained in a small metal package approx. 8 mm in diameter and 4 mm in height [34,35]. The sensor communicated with a microprocessor using I2C communications protocol (Figure 3). The sensor was fixed with a special holding device approx. 0.01 m above the fruit surface (Figure 2 and Figure 4). The MLX 90614 sensor had a built-in emissivity compensation function that allowed the user to change the emissivity coefficient without recalibration. When determining the temperature, a fruit surface emissivity of 0.95 was considered [36] (pp. 363–368). As the emissivity of water is very similar to that of the apple surface, the possibility of condensation affecting the measurement results was eliminated [37]. The sensor had a measuring range of −20 to 120 °C and a very short response time of less than 1 s [38].

Both air temperature and relative humidity (RH) were measured using digital combination sensors (SHT35; Sensirion, Staefa, Switzerland). Owing to the very small dimensions of these sensors, rapid changes in the air parameters could also be recorded. The sensor was placed in a highly permeable plastic housing to prevent direct contact with the fruit surface (Figure 4).

The dew point of the air was determined from the measured values of the air temperature and RH according to the known laws of psychrometry [39,40,41]. For this purpose, the partial pressure of water vapor was first calculated from the air temperature and RH. At the dew point temperature (RH = 100%), the partial pressure corresponded to the partial pressure of the water vapor at saturation. The dew point temperature was determined via nonlinear regression analysis using Table-Curve Tc2D (Systat Software Inc., San Jose, CA, USA) based on the relationship between the partial pressure of water vapor at saturation and temperature. The values for the properties of water vapor from ref. [42] were used as base data for the calculation.

#### 2.4.2. Measurement of Condensate Retention Time with a Wetness Sensor

Resistive wetness sensors were used to measure the total retention time of condensed water on the fruit surfaces [21,43]. Sensors were positioned in the equatorial region of the measuring apple’s surface (Figure 2).

The in-house developed measuring system consisted of the two electrodes of the resistive wetness sensor and an operational amplifier (LM 358, Texas Instruments, Dallas, TX, USA), which were connected to the analog input of the microcontroller acting as master (Figure 3). Condensation on tomato and melon fruit surfaces was previously measured during several hours using copper sensors based on electrical resistance [31,44]. To avoid oxidation of the sensor material during long-term measurement, electrodes made of titanium film were used in this study.

Both the structure and material properties of the electrodes were selected so that full contact with the curved fruit surface was guaranteed. Different structures were, therefore, tested in preliminary studies (Figure 4). With approximately the same length of the contact lines between the electrodes and the apple surface, no significant differences were found for the structures with regard to condensation retention time.

The actual measuring element consisted of an electrically conductive, self-adhesive titanium foil (MOS Equipment, Santa Barbara, CA, USA). The thickness of the tape was approx. 0.1 mm. The electrodes were designed using Silhouette Studio software, cut out with a Silhouette Cameo cutting plotter (Silhouette America Inc., Lindon, UT, USA), and then glued to the fruit surface by hand. Wire connection of the electrodes to the operational amplifier was enabled by an electrically conductive adhesive wire glue (Anders Products, Melrose, MA, USA).

The measuring unit detected wetness from a reduction in electrical resistance in the Megohms range between the electrodes. Depending on the setting and performance parameters of the operational amplifier, this resulted in a dimensionless value of 710 for a dry surface. When condensate began to accumulate, this value dropped down into a range below 200, corresponding to the maximum possible amount of condensate.

The total retention time of condensed water on the fruit surface could be determined from the measured signal history. Whenever the reading was below the dry surface value, water was present on the fruit’s surface. The direction of the signal sequence provided information about the specific condensation phase. A decreasing signal indicated active condensation, whereas an increasing signal indicated evaporation. In principle, the active condensation phase (Tsurf < Tdew) should be in the range of the measured dew point undershoot (Section 2.4.1). Slight time shifts could occur due to the local arrangement of the sensors on apples and the specific prevailing heat and mass transfer conditions.

A preliminary test was carried out to obtain more detailed insight into the condensation process on the sensor surface. For this purpose, a thermal imaging camera (ThermoCAM^®^ HD 600; Infratec, Dresden, Germany) was used to record the surface temperature distribution during the re-warming of an apple (at 20 °C with free convection) after previous cooling. Thermal images (768 × 576 pixels; <0.03 K thermal resolution) were recorded at 2 min intervals over the entire period of re-warming.

As a surfactant, a household washing-up liquid (Palmolive original; Colgate-Palmolive, Hamburg, Germany) containing 5–15% anionic tensids and less than 5% non-ionic tensids was applied before starting the measurement. The surface active agent (50% dissolved in water) reduced the surface tension of the water droplets on the waxy fruit surface and the interface between the fruit and the electrically conductive electrodes. The latter was of particular importance because geometric discontinuities must be overcome [45] to create a closed circuit and signal stability. The wetting agent was applied with a brush to the electrode’s entire surface, the apple surface between the electrodes, and the adjacent edge areas of the sensor electrodes a few hours before the start of the experiment. The fruit surface had to be dry so that there was no conductive connection between the electrodes, and this connection could only be established if there was condensation on the surface. Preliminary tests were performed to estimate the appropriate concentration of the surfactant and the associated drying time. This was particularly important for long-term experiments under frequently changing external climate conditions (condensation/atmospheric evaporation).

#### 2.4.3. Calibration Procedures

All temperature sensors were calibrated for the temperature range of interest (0 to 10 °C) using an EP21 temperature calibrator (SIKA, Kaufungen, Germany). The calibration results were applied to adjust the measuring values via calculation software.

Special attention was required for calibrating the sensors due to the measurement requirements in the case of high air humidity. Five-point calibration procedures were established at three different temperatures (1, 5, and 10 °C) to calibrate the sensor in the high humidity range between 80 and 100% RH. High humidity was achieved using distilled water (=100% RH) and four unsaturated salt solutions (sodium chloride) of different molalities (0.6, 2.0, 3.0, and 5.0 mol kg^−1^) [46]. These molalities corresponded to RH values of 98.1, 93.2, 89.5, and 81.2%, respectively. The equation for the correction describing the correlation between the measured actual value and the reference values of air humidity and temperature was determined by non-linear multiple regression analysis using TableCurve 3D (Systat Software Inc., San Jose, CA, USA).

## 3. Results

The key finding of this study was the identification of recurrent water vapor condensation on apple surfaces using two different measurement techniques during CA storage in the supply air area.

Figure 5 shows the condensate measurement of the wetness sensor with thermal images during and after the re-warming of an apple surface. The electrical circuit was closed by a mixed form of condensation (Figure 5A). Obviously, contact was made by droplets (small dark blue dots) and film (light blue areas on yellow-green background), and the sensor signal indicated condensate in contrast to the sensor element on the dry fruit surface (Figure 5B). Both were visible due to the different emissivity values of water (0.95) and electrically conductive sensor elements. The emissivity of non-polished metal surfaces is usually in the range of 0.1–0.4 [47].

At the measuring point examined in more detail in this study (Figure 1), air velocities of 0.04 m s^−1^ were measured in the vicinity of the apples and 0.2 m s^−1^ in the free airflow in the horizontal gap between the stacked bins of tiers 3 and 4 (Figure 1). The selected measuring point did not represent the measuring point with the highest air velocity.

During the study, a normal refrigeration system cycle (not influenced by the defrosting regime), consisting of cooling and subsequent re-warming phases, lasted approx. 2 h 30 min. Cooling with the refrigeration system in operation took approx. 15 min and re-warming took the remaining 2 h 15 min. The total cycle time could become longer or shorter with changes in the heat load (outside climate, fruit respiration). At the same time, the time relations within a cycle could shift, so that corresponding measurements over a few days or weeks always represented only a snapshot.

Based on the control regime applied, three different scenarios were observed at irregular intervals, which, among other factors, affected the retention time and intensity of condensation on the fruit surfaces.

### 3.1. Cooling Cycle without Interruption by Defrosting during the Re-Warming Phase (Cycle 1)

When the refrigeration system and ventilators were switched on, the cold air temperature at the measuring location (in the area of the apples) dropped very quickly, by more than 1 K. The temperature drop was combined with a decrease in up to 10% relative humidity due to the precipitation of water vapor on the evaporator (Figure 6A). Due to the previous cooling cycle, there was still condensed water on the surface of the fruit during the evaporation phase, as indicated by the signal of the wetness sensor (Figure 6B).

After the lower temperature setpoint was reached, the cooling machine switched off, and the fans continued to run for another 10 min. As a result, both the air temperature and relative humidity at the measuring point (upper fruit layer in the bin, tier 3) initially rose comparatively quickly. Meanwhile, the apple surfaces had dried due to intensive mass transfer. The following period (ventilators off for 40 min) was characterized by a moderate increase in air temperature and relative humidity at the measuring location (Figure 6A,B). This also increased the air dew point, which was initially even lower than the temperature of the fruit surface.

The subsequent activation of the fans (for 20 min) resulted in an intensification of heat and mass transfer combined with a rapid increase in relative humidity (approx. 2%) and a moderate increase in temperature. This briefly led to the first drop below the dew point with only slight signs of condensation. In the subsequent resting phase (fans off for 40 min), the values remained without falling below the dew point, because the fruit surface temperature continued to rise slightly.

The following ventilation phase (20 min) was characterized by active condensation when the air temperature continued to rise and the dew point was slightly lower, which lasted until the upper temperature setpoint was reached (switching on the refrigeration system).

During the study, six complete refrigeration cycles per day were observed without interruption of the re-warming phase by defrosting cycles.

### 3.2. Cooling Cycle Interrupted by Defrosting in the Re-Warming Phase (Cycle 2)

When the refrigeration machine was switched on, the cooling phase proceeded as described above. The same applied to the beginning of the re-warming phase until the defrosting phase began. In the example shown (Figure 7A,B), this phase started while the fans were also in operation. Consequently, the running time of the fans increased (10 to 15 min) in comparison to the Cycle 1 without interruption during the re-warming phase (Figure 6).

Switching on the fans resulted in an intensification of mass transfer associated with an increase in relative humidity (~2%) combined with a moderate increase in temperature. When the temperature dropped below the dew point, the active phase of condensation began. The condensation phase was only slightly dampened by the 10 min rest period after the defrosting process (ventilation off) and then intensified again with the start of the next regular fan operation until the upper temperature setpoint was reached (cooling on).

We observed three cooling cycles per day with interruption of the re-warming phase by the defrosting process.

### 3.3. Refrigeration Machine Cycle, Superimposed by Defrosting during the Cooling Phase (Cycle 3)

Another scenario was characterized by the defrosting cycle starting exactly when the refrigeration system was in operation (Figure 8A,B). As a result, the cooling unit was switched off before the air temperature reached the lower set point.

Resulting from the previous refrigeration machine cycle, condensed water was present on the fruit surface during the active condensation phase (dew point undershoot). With the cooling unit switched on, both the air temperature and relative humidity, and thus the dew point temperature, decreased as a function of the operating time of the refrigeration machine. However, the short run time of the refrigeration machine was not sufficient to dry the apple surface completely. In the following defrosting process (30 + 10 min) and subsequent re-warming phase without fan operation, the dew point was predominantly undershot until the following refrigeration machine cycle, and thus active condensation was recorded.

The wetness sensor indicated condensation on the fruit surface over the entire refrigeration machine cycle (Figure 8B). The fruit surface first started to dry with the onset of the next cooling phase. The total duration of the refrigeration machine cycle after interruption of the cooling phase was shorter than the other cycles because the cooling phase was interrupted prematurely and the fans ran longer, resulting in more intense heat and mass transfer.

We observed two cooling cycles per day with interruption of the cooling phase by the defrosting process.

### 3.4. Comparison of the Cooling Cycle Types

Table 1 contains a comparison of the three types of refrigeration cycles of the cooling operation, as described above.

The duration of a refrigeration machine cycle was the longest in the scenario without interruption of the re-warming phase by a defrosting process (Cycle 1). These results likely occurred due to the shorter fan run time and the associated lower intensification of mass transfer. Defrosting during the re-warming phase (Cycle 2) shortened the cycle by 15% compared to Cycle 1. When the cooling phase was interrupted by a defrosting process (Cycle 3), the refrigeration machine cycle was considerably shorter (Table 1). The fans operated for 30 min during the defrosting process immediately after the refrigeration machine was switched off, thus enabling the air temperature to rise more quickly during Cycle 3.

Similar conditions were observed for the total retention time of condensation on the fruit surface. There were different retention times of condensed water on the fruit surfaces at the selected measuring locations in all three refrigeration machine cycles. The retention time of the condensate was lowest during Cycle 1, which was 33.4% of the duration of a refrigeration machine cycle (Table 1). During Cycle 2, the proportion was almost 10% higher, at 42.2%. During Cycle 3, water condensate was present on the surface over the entire cycle (100%), a result that was also observed during the storage of pears [48]. The interruption scenarios (Cycle 2 and 3) also showed a significant increase in the duration of the active condensation phase compared to evaporation, which could be attributed to the longer fan running times (ventilation on).

A comparison of the time periods for the measured dew point undershoot with the relevant values measured with the wetness sensor led to the following findings. In the case of continuous refrigeration machine cycles (Cycle 1; 6 per day) and the cycle with an interruption during the re-warming phase (Cycle 2; 3 per day), the duration of dew point undershoot was higher than the time of active condensation indicated by the wetness sensor (Figure 6 and Figure 7). The main reason for this was due to the locally different heat and mass transfer conditions. Although the measurement locations on the circumference of an apple were only approx. 10 cm apart, strongly divergent inflow conditions cannot be ruled out, especially in the phases with forced airflow. Experiments under laboratory conditions have shown that partial areas with and without condensed water can exist simultaneously on the apple surface [21]. We assume that the wetness sensor itself had a minor influence on the recording of condensation duration due to the very low thickness of the titanium foil and the high thermal conductivity of titanium compared to the fruit. The surfactant applied for signal stability of the wetness sensor might have become diluted during storage due to partial evaporation and from condensation droplets rolling off the curved surfaces of the apple. Therefore, the condensation intensity, which was not determined in this study, may have been affected. An influence on the retention time of the condensate can be excluded.

In contrast, the mean period of dewpoint undershoot for Cycle 3 with interrupted cooling phases (2 per day) was shorter than the time of active condensation. This was primarily due to the running time of the refrigeration machine before shutdown by the defrost control, which was very different among the three cycles and recognizable from the comparatively large standard deviations.

At the selected measuring point in the upper layer of apples of tier 3, the condensation potential was relatively low, with a maximum of 0.25 K below the dew point. However, from the results of the wetness sensors, it can be concluded that the fruit in the supply air area with high air velocity and strong temperature fluctuations had wet surfaces for more than one third of the storage time.

The question of whether the condensation process tends to have a positive or negative effect on mass loss and overall produce quality must also be clarified in subsequent investigations. Investigations performed in previous years in the same cold storage rooms with ‘Jonagold’ apples showed similar mass losses, reduced temperature fluctuations, and slightly higher air temperatures in the supply air area with high air velocity than in areas with low air velocity [7]. The benefits of condensation, such as reduced transpiration, must be weighed against its negative consequences, including increased microbial growth. The condensation of moisture on the commodity is likely a more significant factor in promoting decay than the humidity level of the surrounding air [20] (pp. 13–18). A reduction in condensation could be achieved by changing the control of the cooling component operation. Defrosting during cooling operation (Cycle 3) should be avoided and reduced ventilation intensity during the re-warming phase of cooling cycles could be considered. A reduction of the usual fan power by >50% still ensured air movement between fruit at the top and bottom positions of the bins in a common industrial storage room [6].

The wetness sensors provided usable measurement signals for approx. four weeks under the given conditions, with multiple daily changes of moistened and dry fruit surfaces. Based on the research results, it can be clearly seen that the monitoring system was capable of real-time measurements, easily collected a large amount of data with high resolution, and was robust even under changing condensation conditions.

## 4. Conclusions and Outlook

This study was the first to demonstrate condensation on fruit surfaces under real conditions in an apple CA storage room. Using the example of a measuring point in the upper layer of apples in a large bin, it was possible to provide evidence that frequent drops below the dew point occur, thus leading to condensation on apple surfaces.

The condensation process was detected using two independent measuring methods. Furthermore, the study clarified how long the effect of a surfactant lasts under the given conditions, with constant alternation of condensation and evaporation and associated dilution of the wetting agent, and how long the wetness sensor can generate sufficient signals. With regard to data acquisition, it should be noted that the advantages of an IoT-based system, such as low price, easy modification, and expandability of the measurement configuration with additional components, are also offset by disadvantages such as increased inputs for preparing the sensors for use, e.g., calibration, preparation, and post-processing of the measurement results (filters, plausibility checks, etc.).

The influence on condensation of the refrigeration operating system with its components was shown, including the cooling unit, fans, and defrost control. Defrost control had a clear influence on the total retention time of the condensed water whenever the defrosting process interrupted the cooling phase. The timed use of the fans led to an intensification of heat and mass transfer, thereby having a noticeable influence on the condensation process.

This new finding raises a whole series of unanswered questions. The results presented here only apply to the selected measuring location in the supply air area with larger air temperature fluctuations caused by the cooling unit. Similar results can be expected for the entire supply air area with more or less strong effects on the retention time of the condensed water on the fruit surfaces. The results cannot be used to draw conclusions about condensation phenomena in areas with lower temperature fluctuations, lower air velocities in the storage room, and within large bins. The same statement also applies to temporal changes in the cooling load (cooling down phase of the apples, changing metabolic processes, external climatic conditions) with respect to condensation phenomena since the results presented here covered only a period of approx. four weeks.

With regard to the influence of the mode of operation on condensation processes, further investigation is required. This also applies to special sections of storage, such as the time immediately after the apples have been stored during cooling of the fruit and storage phases with active additional heating during the defrosting process. Further investigation should also include the spatial distribution of areas with and without condensation in the cold room (supply air and backflow areas) and the resulting different post-storage behavior of the fruit.

The influence of the constant alternation of dew formation and evaporation on apple quality and post-storage behavior (shelf life) should be the subject of a subsequent investigation. Only then (also taking into account the apples stored in the air backflow area) can the use of the individual components of the cooling system be improved in relation to the energy efficiency of the overall process.

The results of the present study show the possibility for further energy savings through operation of the cooling system including its components, independent of the condensation problem. This applies in particular to the selection and use of fans, which have a strong influence on the intensity of heat and mass transfer. However, the effect on all fruit in the supply and air backflow areas must be taken into account.

## Figures and Tables

**Figure 1 sensors-23-01739-f001:**
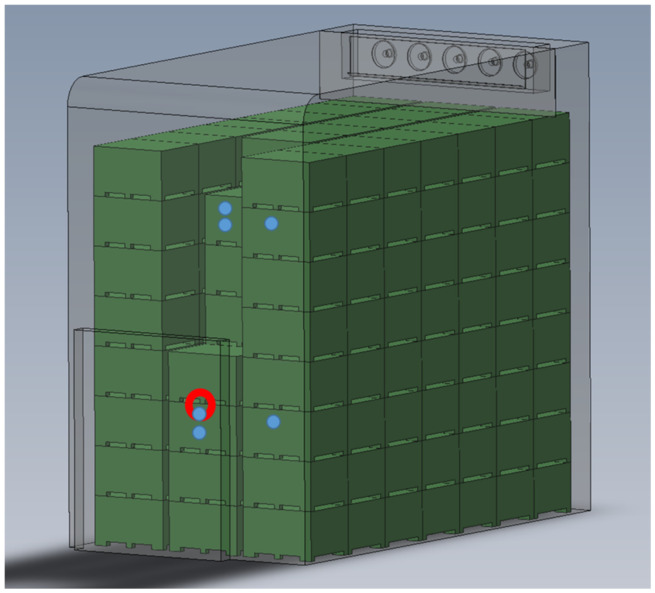
Overall view of the cold room with storage bins, cooling unit (evaporator and fans), all bins measured (blue dots), and selected measuring point (red circle) in the supply air area.

**Figure 2 sensors-23-01739-f002:**
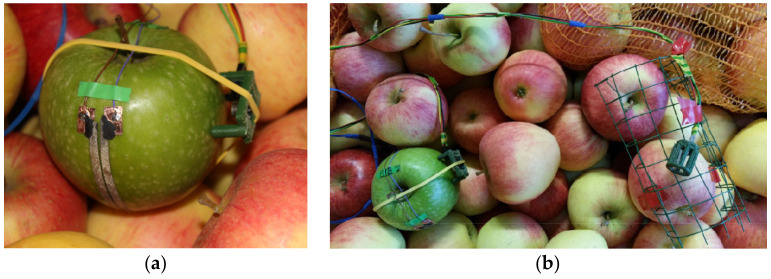
Wetness sensor and infrared sensor at the measuring apple (**a**) and grouping of the sensors in the upper layer of the apple bin (bottom left, wetness sensor and infrared temperature sensor attached to the same apple; bottom right, air temperature and relative humidity sensor) (**b**).

**Figure 3 sensors-23-01739-f003:**
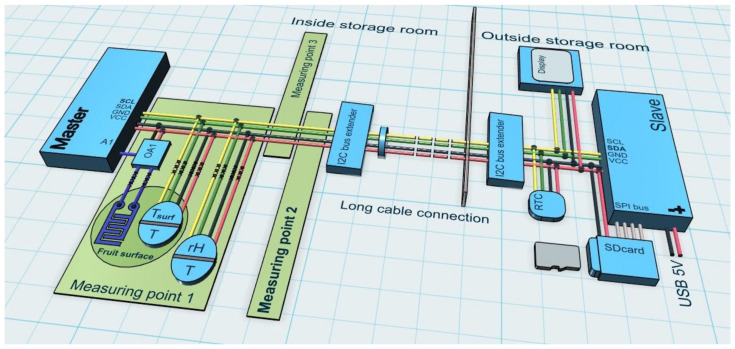
Data acquisition system in master-slave configuration, two digital sensors and one analog sensor with operational amplifier for each of 3 measuring points. I2C: inter-integrated circuit; SCL: serial clock; SDA: serial data; OA1: operational amplifier measuring point 1; VCC: voltage at common collector; GND: ground; A1: analog input pin 1; RTC: real time clock; T: temperature; Tsurf: surface temperature; rH: relative humidity; SPI: serial peripheral interface.

**Figure 4 sensors-23-01739-f004:**
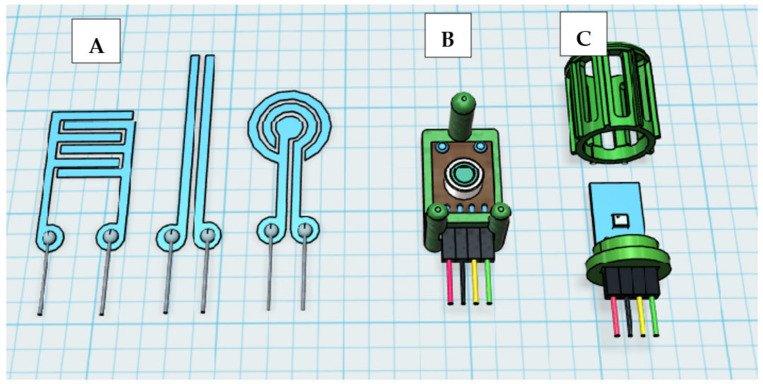
Tested electrode structures of the wetness sensor (**A**), infrared temperature sensor with holding device (**B**) and temperature/humidity sensor with holding device/basket (**C**).

**Figure 5 sensors-23-01739-f005:**
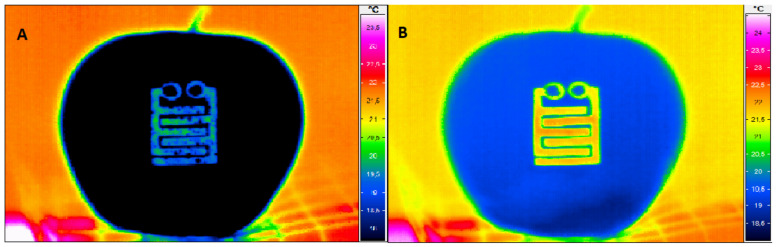
(**A**) Condensate on a cold apple surface during re-warming with free convection: Distribution of condensed water droplets in the sensor film area. (**B**) No sign of condensation in the sensor area after progressive re-warming of a dry apple surface.

**Figure 6 sensors-23-01739-f006:**
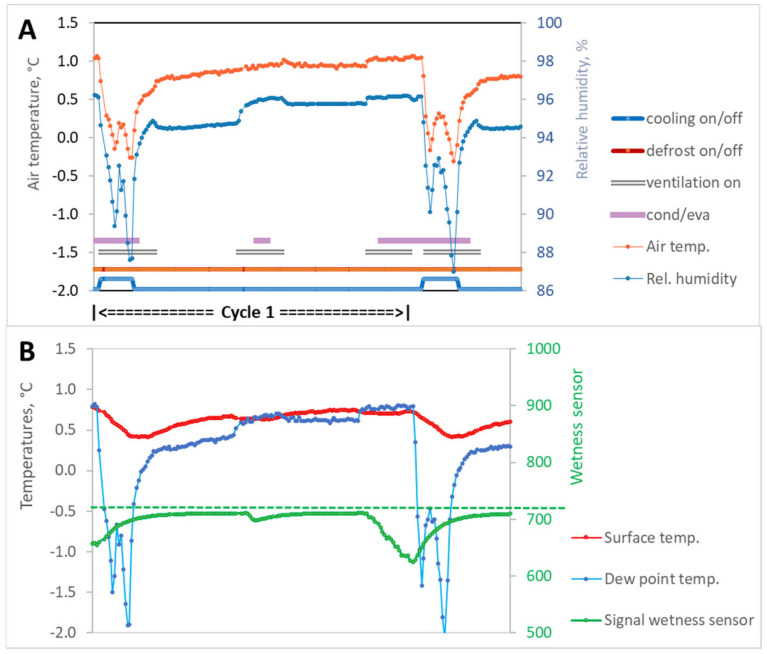
Cycle 1 without defrosting process in the re-warming phase. (**A**) Parameters of the cold air (temperature, relative humidity) adjacent to the produce, operating mode of the components of the cooling system (cooling, defrost, ventilation), and total retention time of condensed water (cond/eva) on the fruit surface. (**B**) Produce surface temperature, dew point temperature of the surrounding air, and signal history of the wetness sensor. The signal history of all sensors shown here covers a period of 3 h and 20 min. All parameters were measured in the top layer of apples in the bin in middle row of tier 3.

**Figure 7 sensors-23-01739-f007:**
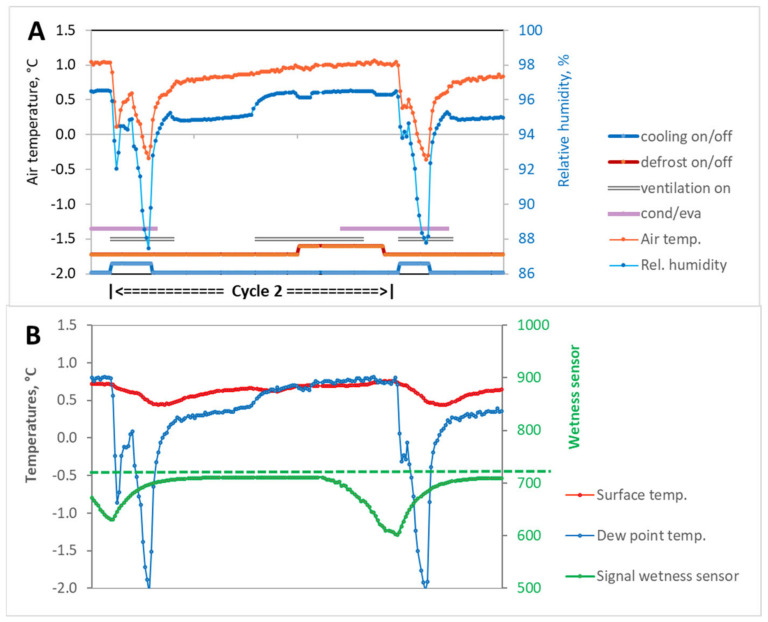
Cycle 2 with defrosting process during re-warming phase. (**A**) Parameters of the cold air (temperature, relative humidity) adjacent to the produce, operating mode of the components of the cooling system (cooling, defrost, ventilation), and total retention time of condensed water (cond/eva) on the fruit surface. (**B**) Produce surface temperature, dew point temperature of the surrounding air, and signal history of the wetness sensor. The signal history of all sensors shown here covers a period of 3 h and 20 min. All parameters were measured in the top layer of apples in the bin in middle row of tier 3.

**Figure 8 sensors-23-01739-f008:**
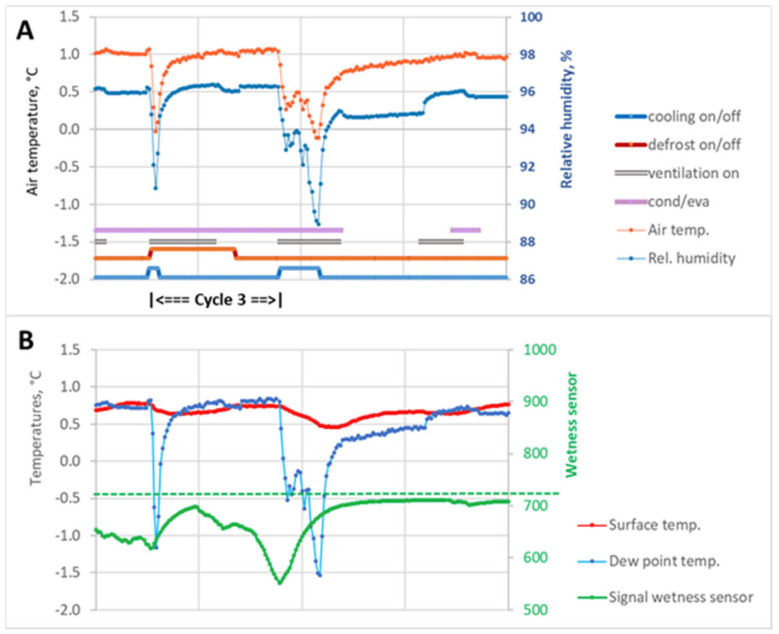
Cycle 3 with interruption of the cooling phase. (**A**) Parameters of the cold air (temperature, relative humidity) adjacent to the produce, operating mode of the components of the cooling system (cooling, defrost, ventilation), and total retention time of condensed water (cond/eva) on the fruit surface. (**B**) Produce surface temperature, dew point temperature of the surrounding air, and signal history of the wetness sensor. The signal history of all sensors shown here covers a period of 3 h and 20 min. All parameters were measured in the top layer of apples in the bin in middle row of tier 3.

**Table 1 sensors-23-01739-t001:** Average total duration of five refrigeration machine cycles and proportional duration of the times with fan operation, total retention time of condensate on the fruit surface, and proportional times for active condensation and evaporation (mean value ± SD, n = 5). Percentages are given in brackets.

	Without Defrosting during Re-Warming Phase (Cycle 1)	Defrosting during Re-Warming Phase (Cycle 2)	Defrosting during Cooling Phase (Cycle 3)
Total cycle time (min)	154.9 ± 7.6	133.3 ± 11.5	55.2 ± 10.3
Air temperature (°C)	0.83 ± 0.28	0.82 ± 0.30	0.94 ± 0.05
Relative humidity (%)	95.00 ± 1.55	94.91 ± 1.79	95.95 ± 0.48
Ventilation on (min)	66.6 (43.0)	70.2 (52.7)	30.7 (55.6)
**Wetness sensor** **:**			
Cond. retention time (min)	51.7 ± 8.9 (33.3)	56.3 ± 7.9 (42.2)	55.2 ± 10.3 (100.0)
Active condensation (min)	24.9 (48.2)	33.0 (56.8)	30.7 (55.6)
Atm. evaporation (min)	26.8 (51.8)	23.3 (41.4)	24.4 (44.2)
**Dewpoint measurement** **:**			
Dewpoint undershoot (min)	33.8 ± 2.2 (21.8)	45.4 ± 6.6 (34.0)	23.0 ± 12.7 (41.7)

## Data Availability

Not applicable.

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
