# Peer review of "Measurement of Water Vapor Condensation on Apple Surfaces during Controlled Atmosphere Storage"

_sensors, 2023, doi:10.3390/s23031739_

Round 1

Reviewer 1 Report

 In this paper, two independent measuring methods are used to detect the condensation process and the results proved the phenomenon of condensation on fruit surfaces during the controlled atmosphere storage of apples. Although the theme of this research is significant and the results are interesting, some following issues should be considered to improve the quality of this paper. I would like to recommend its publication after minor revisions.

1. All abbreviations should be explained in the text, which is helpful for readers in other fields.

2. This paper uses temperature and humidity sensors to demonstrate the phenomenon of condensation on the fruit surface during controlled atmosphere storage of apples, which is unfavorable for fruit storage. Can the author give some advice on reducing condensation on the fruit surface.

3. There are many measurement points in this experiment, and some key data can be given statistical graphs in this paper, which can make the experimental results more convincing.

4. The green apples selected for testing for Figure 2 are not similar to other apples, whether this will affect the accuracy of the experimental results.

5. In the experiment, the surface of the tested apples was brushed with surfactant. Please comment whether this has any effect on the results.

Author Response

Thank you very much for your  comments and suggestions. Here you find our answers and changes in the revised manuscript.

All abbreviations should be explained in the text, which is helpful for readers in other fields.

The following explanations of abbreviations were added in the text:

L 30: Controlled Atmosphere (CA)

L 170: Air speed logger (ASL)

L188 Secure Digital (SD)

L 252 relative humidity (RH)

This paper uses temperature and humidity sensors to demonstrate the phenomenon of condensation on the fruit surface during controlled atmosphere storage of apples, which is unfavorable for fruit storage. Can the author give some advice on reducing condensation on the fruit surface.

Whether condensation on the fruit during storage at low storage temperatures has a negative effect is still unclear, as condensation prevents water loss (Line 65-67). Condensation occurs only during the re-warming phase due to enhanced increase of humidity in the vicinity of the product when the fans are running.

Approaches to avoid condensation are:

-reduction of ventilation intensity (fan power), maintenance of air movement between fruit in bins at upper and lower positions in the warehouse was shown in a previous study [6] (Praeger et al. 2021)

-Targeted timing of the defrosting times to avoid overlapping of the cooling phase by defrosting.

Please see added advice in the text chapter Results: L 543-549

There are many measurement points in this experiment, and some key data can be given statistical graphs in this paper, which can make the experimental results more convincing.

This paper presents for the first time the detection of condensation processes on fruit surfaces during real long-time fruit storage. The paper focusses on the presentation of the sensor technology for 2 independent measurement methods, data transmission in the apple store and the relationship between the condensation phases and the cooling component operation.

For the analysis in this publication, we consider it appropriate to present a representative measuring point in the supply air area of the storage room, taking into account 5 repetitions each for the 3 different types of cooling cycles.

Further data analyses of the remaining measuring points are planned in order to analyse the influence of the measuring position.

The green apples selected for testing for Figure 2 are not similar to other apples, whether this will affect the accuracy of the experimental results.

An apple variety ‘Granny Smith’ was used as a measuring apple due to the long shelf life. Condensation is influenced by thermal product properties which are very similar in different apples due to the similar high water content (about 85%). Differences in the peel surface texture has probably only a marginal effect due to the use of the sensor film that rests on the peel as long as a complete contact to the sensor film is ensured. The focus of the investigation was not on the influence of the variety, but on the detection of condensation on apple surfaces depending on the operation of the cooling components.

Please see added text Line 160-161

In the experiment, the surface of the tested apples was brushed with surfactant. Please comment whether this has any effect on the results.

The surfactant ensures that the surface tension of the water is reduced. This provides (at least in part) a film that provides a more stable signal. In laboratory tests, the appropriate concentration was determined in order to allow qualitative detection of condensate formation (duration of condensate on the apple surface) with the wetness sensor. On dry apple surface coated with the surfactant, the sensor signal does not indicate condensation. The surfactant is diluted over time by the constant alternation of defrosting and drying (rolling of droplets during defrosting + evaporation). This led to the signal not being measurable after about 3 weeks over the period of several hours. It could affect the intensity, not the length of retention time of condensate.

Please see added text Line 519-524

Reviewer 2 Report

In my concern, the current organization of this paper makes it not suitable for publication in sensors, since I cannot see its scientific or technique significance clearly. It's a complete agriculture monitoring process and I do accept its significance in this field. As a reader of Sensors, we want to see the novelty in either of the sensor design, sensing method, or sensing data processing, etc. However, this work doesn’t organize in the way of a scientific work. For example, in Introduction, readers want to see why we choose the topic and why this method is in a high novelty, and why it is better than previous method. In the Methods section, we want to see a more focused  description of a novel method rather than a complete process. The current writing is a little bit disordered that readers cannot get the focusing point.

Author Response

Thank you very much for the review and your comments and suggestions.

Here you find our answers to the comments and the revised manuscript. We improved the presentation of the functionality and application of a new wetness sensor and indirect determination of condensation processes during long-term storage of apples, that are shown in this study for the first time. Also relevant literature was added.

The entire text has been revised to improve the expression and grammar in English and to focus on essential points.

The novelty and importance of the topic was pointed out in the introduction ( Line 68-79, 97-103).

Additions have been made to explain the use and functionality of the sensors.

On the one hand, details have been added to justify the choice of measurement methods, both of which are new for use in cold storage of fruit. (Line 278-282)

On the other hand, in-depth information on the sensor technology (condensate forming and dew point undershoot) and on the conditions of use were incorporated. (Line 240-250, 303-308, 321-329, 347-363)

It should also be pointed out once again that the findings obtained with the new sensor combination can have a far-reaching influence on subsequent investigations.

In our opinion, these measurements are important for the development of a climate control in storages with optimised energy efficiency and quality monitoring.

Round 2

Reviewer 2 Report

The revisions given by the authors are sufficient, and I belive the final version manuscript are qualified for publication.